# Pressurized Intraperitoneal Aerosol Chemotherapy (PIPAC) with Oxaliplatin, Cisplatin, and Doxorubicin in Patients with Peritoneal Carcinomatosis: An Open-Label, Single-Arm, Phase II Clinical Trial

**DOI:** 10.3390/biomedicines8050102

**Published:** 2020-04-30

**Authors:** Michele De Simone, Marco Vaira, Monica Argenziano, Paola Berchialla, Alberto Pisacane, Armando Cinquegrana, Roberta Cavalli, Alice Borsano, Manuela Robella

**Affiliations:** 1Candiolo Cancer Institute, Unit of Surgical Oncology, FPO-IRCCS, 10060 Candiolo, Italy; michele.desimone@ircc.it (M.D.S.); marco.vaira@ircc.it (M.V.); armando.cinquegrana@ircc.it (A.C.); alice.borsano@ircc.it (A.B.); 2Department of Drug Science and Technology, University of Turin, 10125 Turin, Italy; monica.argenziano@unito.it (M.A.); roberta.cavalli@unito.it (R.C.); 3Department of Clinical and Biological Sciences, University of Turin, 10126 Turin, Italy; paola.berchialla@unito.it; 4Candiolo Cancer Institute, Pathology Unit, FPO-IRCCS, 10060 Candiolo, Italy; alberto.pisacane@ircc.it

**Keywords:** peritoneal carcinomatosis, chemotherapy, aerosol, cancer

## Abstract

Pressurized intraperitoneal aerosol chemotherapy (PIPAC) is an innovative approach for treating peritoneal carcinomatosis that applies chemotherapeutic drugs into the peritoneal cavity as an under-pressure airflow. It improves local bioavailability of cytostatic drugs as compared to conventional intraperitoneal chemotherapy. The aim of this study is to prove feasibility, efficacy and safety of this new treatment. Patients included in the analysis underwent at least two single port PIPAC procedures; drugs used were Oxaliplatin for colorectal cancers and Cisplatin + Doxorubicin for ovarian, gastric, and primary peritoneal cancers. The primary endpoint was the Disease Control Rate according to the RECIST (Response Evaluation Criteria in Solid Tumors) criteria. Secondary significant endpoints were overall and progression free survival, tumor regression on histology, and quality of life. Safety and tolerability were assessed according to the Common Terminology Criteria for Adverse Events 4. Sixty-three patients were enrolled in this trial. Forty patients (100 PIPAC) were eligible for analysis. Twenty patients were undergoing systemic chemotherapy. Fourteen patients reported an objective response (35%). Median overall survival was 18.1 months; median progression-free survival was 7.4 months. Minor morbidity was observed in seven procedures. Grade 3 complications occurred in two patients, and grade 4 in one patient submitted to reoperation. Single-port PIPAC is feasible, safe, and easy to perform. The combined treatment based on systemic chemotherapy and PIPAC does not induce significant hepatic and renal toxicity and can be considered a valid therapeutic option in patients with advanced peritoneal disease. Further studies on the use of PIPAC alone, possibly with different drug dosages, may define the real effectiveness of the procedure.

## 1. Introduction

Peritoneal carcinomatosis (PC) is a common manifestation of advanced gynecological and gastrointestinal malignancies, still associated to bad prognosis in the absence of an aggressive multimodal therapy. Although the combined treatment of cytoreductive surgery (CRS) and hyperthermic intraperitoneal chemotherapy (HIPEC) has been increasingly proposed as a curative treatment, the therapy of peritoneal metastasis remains largely palliative.

CRS and HIPEC represent the gold standard treatment for pseudomyxoma peritonei (PMP) [1,2] and peritoneal mesothelioma (DMPM) [3,4], and the curative approach for highly selected patients affected by ovarian, colorectal, and gastric cancers [5,6,7].

For non-resectable PC, palliative systemic chemotherapy (sCT) is still, nowadays, often the standard of care. Some cheering improvement in survival was recorded in PC from colonic cancer, in which median survival raised from six to 24 months by novel drug agents [8]. In other PC, such as from gastric cancer, results are not so encouraging, with median survival rates of 6–10 months [9]. In ovarian cancer, intravenous chemotherapy in various combinations and sequences is the mainstay of recurrence treatment; these regimens achieve median overall survival rates ranging from 18 to five months [10]. It is remarkable that the intraperitoneal availability of drugs by sCT is low. Consequentially, sCT is often inefficient in bulky disease. Furthermore, the cumulative toxicity of systemic repeated sCT regimens is responsible for the progressive decrease of patients’ compliance to therapy.

Pressurized intraperitoneal aerosol chemotherapy (PIPAC) is an innovative intraperitoneal chemotherapy approach for peritoneal carcinomatosis that could overcome the limitation of IPC (IntraPeritoneal Chemotherapy) [11]. This new technique is based on the laparoscopic pressure that is able to enhance drug distribution and uptake [12,13,14]. Better distribution and penetration are demonstrated in the animal model compared to liquid solutions [15,16].

On the basis of these considerations, we tested PIPAC within the framework of a single center, open label, non-randomized, single-arm, repeated single dose study to explore the efficacy, safety, and DCR (Disease Control Rate) of cisplatin, oxaliplatin, and doxorubicin in patients with PC from ovarian, gastric, and colorectal cancers and in primary cancers of the peritoneum.

## 2. Materials and Methods

### 2.1. Study Design

The target subject population is composed of patients with ovarian and colorectal cancers reporting relapse, or disease progression with peritoneal carcinomatosis after at least two lines of previous IV standard chemotherapy; and patients with recurrent gastric cancer, PMP, and primary peritoneal tumors, reporting disease relapse or progression with peritoneal carcinomatosis, after at least one line of previous IV standard chemotherapy. Patients had to be ineligible for cytoreductive surgery +/- HIPEC, and not tolerate or refuse systemic chemotherapy.

Specifically, patients were eligible if they had clinical and pathological confirmation of peritoneal carcinomatosis from gastric, colorectal, and ovarian cancers or primary peritoneal tumors; were aged between 18 and 78 years; had good performance status (ECOG ≤ 2), blood and electrolytic counts, and liver, renal, and cardiopulmonary function parameters within 10% of the normal range; and a tumor mass present on CT-scan, in order to allow tumor response assessment with RECIST-criteria. All patients signed an informed consent form.

Any of the following are regarded as an exclusion criterion from the study: extra-abdominal metastatic disease, with the exception of isolated pleural carcinomatosis; bowel obstruction; severe renal impairment; myelosuppression; severe hepatic impairment; severe myocardial insufficiency; recent myocardial infarction; severe arrhythmias; immunocompromised patients; or pregnancy. Progressive ascites was not an exclusion criterion.

The PI-CaP study was approved by the Italian drug agency (AIFA; Agenzia Italiana del FArmaco) and Istituto Superiore di Sanità. The study was registered with ClinicalTrials.gov, number NCT02604784 and EudraCT number 2015-000866-72 (approved on 30 April 2015). 

The primary endpoint of the study was the DCR according to RECIST criteria (version 1.1) after two cycles of PIPAC. DCR was defined as the total number of patients with complete radiological response, partial response, and stable disease. A baseline abdominal CT scan was required ≤4 weeks before the first PIPAC procedure and repeated after the second PIPAC cycle. A further CT scan was not performed, in the case of clinical progressive disease. Secondary outcomes were the clinical tumor response to therapy, using ^18^F-FDG PET according to PERCIST (Positron Emission Tomography Response Criteria in Solid Tumors) criteria (version 1.1); overall survival (OS); progression-free survival (PFS) according to RECIST criteria (version 1.1) after two or three cycles of PIPAC; degree of histological regression assessed by pathological review; and evaluation of specific circulating tumor markers before and after the first, second, and third PIPAC.

Quality of life was evaluated according to the Short Form 36 Health Survey (SF-36), before the beginning of the treatment and after each PIPAC procedure.

Safety and tolerability were assessed by the collection of adverse events, according to the Common Terminology Criteria for Adverse Events (CTCAE) 4, including physical examination results and laboratory assessments (chemistry and hematology).

The survival follow-up data were regularly updated by medical examination or telephone calls. The last patient’s last visit (the study required at least 1-year follow-up after the second/third PIPAC procedure) was 30 October 2018. Clinical, laboratory and histopathology data were documented in our hospital information system and reported in a paper clinical record form. The histopathological response (regression grading) was assessed according to the PRGS [17]. The pathologists could compare biopsies taken during previous PIPAC procedure but were blinded to the clinical outcomes.

### 2.2. Operative Procedure

PIPAC procedures were scheduled every 6–8 weeks. Patients undergoing concurrent sCT had a wash-out period of 7–10 days before and after each PIPAC.

PIPAC procedures were performed under general anesthesia. Venous thromboembolism prophylaxis was given the night before surgery using Enoxaparin 4000 IE. A single intravenous injection of cefazolin 2 g was administered about 30 min prior to surgery.

A midline single-port access (Olympus Quadport + platform) was realized and a 12 mm Hg CO_2_ pneumoperitoneum was applied. Explorative laparoscopy was carried out and Peritoneal Cancer Index (PCI) was determined. Peritoneal biopsies were taken for histological examination. Ascites volume was documented and removed. A nebulizer (Capnopen^®^, Capnomed, Villingendorf, Germany) was then connected to an intravenous high-pressure injector (Arterion Mark 7^®^, Medrad Healthcare, Germany) and inserted into the abdomen. The tightness of the abdomen was controlled through a zero flow of CO_2_. A pressurized aerosol containing oxaliplatin at a dose of 92 mg/sm body surface in 150 mL dextrose solution was applied in patients with intestinal peritoneal carcinomatosis; an aerosol containing Cisplatin 7.5 mg/sm body surface in 150 mL NaCl 0.9% + Doxorubicin 1.5 mg/sm body surface in 50 mL NaCl 0.9% was administered in patients with gastric, ovarian, and primary peritoneal cancer. The chemotherapy injections were remote-controlled and no personnel remained in the operating room during the application. The flow rate was 30 mL/min and the maximal upstream pressure was 200 psi. The aerosol was maintained for 30 min at 37 °C and a pressure of 12 mmHg, then exsufflated via a closed line over two sequential micro-particle filters into the air waste system of the hospital. The single-port access was removed. No drainage of the abdomen was applied. The PIPAC procedure was repeated three times every 6–8 weeks.

### 2.3. Statistical Analysis

The primary objective was to evaluate the DCR (proportion of patients achieving a complete radiological response, a partial response and stable disease divided by the total number of patients enrolled) according to the RECIST criteria. We used an optimal two-stage Simon’s design to differentiate between a DCR of 15%, below which the method is not considered clinically useful, and 30%, which is the DCR of clinical interest with type I and II errors of 0.0477 and 0.1994, respectively. Under these assumptions, we calculated that 19 patients had to be enrolled in the first stage. If more than three radiological responses were observed, the study should proceed to a projected accrual of 55 patients. The observation of a total of at least 13 responses will confirm a DCR of 30%. Under these conditions, the probability of early termination because the treatment is not effective is 0.6841. Considering an expected drop-out rate of 15%, the final sample size was established, at 63 patients. The expected DCR must be confirmed by the execution of at least two PIPAC procedures.

Statistical analysis was carried out in the per-protocol population. Demographic data were summarized as frequency with percentage for categorical variables, and median with interquartile range for continuous variables.

Overall survival and progression free survival curves were estimated using the Kaplan–Meier methods and compared using the log-rank test. *p*-values less than 0.05 were considered statistically significant. Analyses were performed using R version 3.6.0 software.

## 3. Results

Between October 2015 and December 2018, 171 single-port PIPAC procedures in 82 patients presenting PC from different primary tumors were performed. Between October 2015 and October 2017, 63 patients were enrolled in the PI-CaP trial. Data were locked in October 2018, when the last follow up of the last patient occurred. Laparoscopic non-access rate was 2.9% (2/67). In two cases, PIPAC procedure were not performed because of bowel obstruction.

Sixty-three patients were submitted to at least one PIPAC procedure (ITT: Intention To Treat population); these patients were included in the safety analysis. The Per-Protocol (PP) population was composed of 40 patients (patients submitted to at least two PIPAC procedures). The preplanned 2° PIPAC was not performed due to progression of disease (*n* = 6), deterioration of general condition (*n* = 9), palliative or rescue surgery execution (*n* = 8). Twenty patients received three PIPACs, 20 patients were not submitted to three PIPAC due to progression of disease (*n* = 12) and general condition deterioration (*n* = 8). Patients characteristics are detailed in Table 1.

The protocol initially envisaged the enrollment of non-candidates for systemic chemotherapy (sCT). As a result of poor accrual problems, patients performing sCT were also included. Therefore, it was possible to split the population into two subgroups: patients undergoing sCT and PIPAC (*n* = 20), and patients undergoing exclusive PIPAC procedure (*n* = 20). Diagram of the patients’ flow through the study is depicted in Figure 1.

The mean duration of the first, second, and third PIPAC procedures were 82 (range 45–148), 80 (range 45–150), and 88 (range 60–135) min, respectively. Mean hospital stay was 2.4 (range 2–6), 2.4 (range 2–8), and 2.6 (range 2–6) for the first, second, and third operation, respectively.

### 3.1. Efficacy

The median Overall Survival (OS) rate of the ITT population was 15 months, with a 1 and 2-yrs OS of 67% and 40%, respectively. There was no statistically significant difference in terms of median OS between the group submitted (15 months) or not (20.6 months) to sCT (*p* = 0.72). Median Progression-Free Survival (PFS) was not reached; the 1 and 2-yrs PFS was 58% and 54%, respectively, without any statistically significant difference with or without sCT (*p* = 0.97).

Median OS of the PP population was 18.1 months, with a 1- and 2-yrs OS of 61% and 40%, respectively. Even in this subgroup, any statistically significant difference is due to sCT (20.6 without sCT vs. 18.1 with sCT (*p* = 0.53)). Median PFS was 7.4 months.

Overall survival and time to progression curves of the PP populations are given in Figure 2 and Figure 3. Outcome curves stratified by pathologies are shown in Figure 4 and Figure 5.

In the PP population, the DCR was 35%: seven patients had a partial response (17.5%), and seven patients a stable disease (17.5%) shown by a Computed Tomography (TC) scan according to RECIST criteria (version 1.1). After two PIPAC procedures, clinical tumor response at 18F-FDG- PET according to PERCIST criteria (version 1.1) was reported in 13 patients (32.5%). We observed a stable disease in six cases, a partial response in seven, a progression disease in 11 (not evaluable in 16 patients). The degree of histological regression was assessed according to PRGS (Peritoneal Regression Grading Score). After two PIPAC procedures, we reported a PRGS of 1, 2, 3, and 4 in six, one, 16, and six patients, respectively; in 10 patients it was not evaluable. After the third PIPAC, a PRGS of 1, 2, 3, and 4 was registered in three, zero, eight, and one patients, respectively. Overall, nine partial histological regressions, 11 disease stabilities, and two progression of diseases were recorded. In some cases, it was not possible to perform biopsies, or the specimens were not adequate for the assessment.

The evaluation of specific circulating tumor markers before and after each PIPAC procedure, considering the several pathologies, were not significant. However, Ca125 for ovarian cancer and DMPM, and CEA and Ca19.9 for gastric and colorectal cancer were taken into consideration. A general tendency to slightly increase was identified after the first procedure, followed by a decrease after the second and the third PIPACs in patients where an objective ORR was found. Figure 6 shows the different circulating tumor markers after each procedure.

### 3.2. Safety and Quality of Life

Safety and tolerability were assessed by collection of adverse events, according to the Common Terminology Criteria for Adverse Events (CTCAE) 4, including physical examination results and laboratory assessments (chemistry and hematology). Complication rate was assessed on the total number of interventions (considering the ITT population).

Overall complication rate was 8.1% (10/123). Major complication rate (CTCAE 3–4) was reported in three cases (2.4%).

The most common complications were abdominal pain, nausea, hyperthermia, wound infection, anemia, and leucopenia. Only one patient was submitted to reoperation because of bleeding, likely due to surgical procedure. Thirty-day postoperative mortality rate was nihil. Renal and hepatic functions were not impaired.

Quality of life was evaluated according to the Short Form 36 (SF-36) health survey before the beginning of the treatment and after each PIPAC procedure during the interval control (IC). The SF-36 consists of eight scaled scores, which are the weighted sums of the questions in their section. Each scale is directly transformed into a 0–100 scale, under the assumption that each question carries equal weight. The lower the score, the more disability; the higher the score, the less disability.

Before first PIPAC, baseline general health was 49, emotional role functioning was 63.5, and bodily pain was 74.5. These sections were measured as 49.5, 63.5, and 71 after the first PIPAC procedure, and 45.5, 64, and 71 after the second one, respectively. Prominent complaints were fatigue, and digestive symptoms, such as nausea and constipation. In Figure 7, the detailed scores of each section are shown. No statistically significant changes were observed.

## 4. Discussion

In this prospective open label phase 2 study, safety of PIPAC procedure was firstly investigated. No postoperative mortality was reported, and only one CTCAE grade 4 surgery-related complication was registered. In two patients, CTCAE grade 3 morbidity was reported: one patient for anemia, the other one to be readmitted because of nausea and abdominal pain. Minor complications were anemia, amylase increase, leukopenia, and abdominal pain. These results were comparable to that reported in two other phase II studies: Struller [18] reported in his analysis on patients with peritoneal carcinomatosis of gastric origin three grade 3 toxicities (12%), while Tempfer [19] described eight grade 3 toxicities in 53 patients with advanced ovarian cancer (15.1%). Our results are also in accordance with a recent review on 22 studies (data analysis on 1197 reported adverse events of CTCAE grade 1, 2, 3, 4, and 5 in 45%, 14%, 7%, 0.8%, and 1.6%, respectively [20]). This study also demonstrated the safety and the feasibility of the combined treatment by systemic chemotherapy and PIPAC. There was no difference in terms of complications between the two treatment groups and no cumulative toxicity was reported. Median hospital stay was 3.2 days.

Since PIPAC is still primarily a palliative treatment, quality of life is a significant aspect. Evaluations were made according to the SF-36 health survey before the beginning of the treatment and after each PIPAC procedure during the interval control. Eight scaled scores were investigated (the lower the score, the more disability). Baseline general health was 49, emotional role functioning was 63.5, and bodily pain was 74.5. These sections were measured as 49.5, 63.5, and 71 after the first PIPAC procedure, and 45.5, 64, and 71 after the second one, respectively (Figure 6). The most common complaints were nausea, constipation, and fatigue. No statistically significant changes were observed, neither during subsequent PIPACs, nor between the two treatment groups. This aspect suggests that PIPAC can be considered an attractive option in salvage situations in fragile patients, already subjected to multiple systemic treatments.

In terms of treatment response, we reported a DCR of 35%. The hypothesis of this study was that PIPAC could induce a DCR in ≥30% patients, as assessed by RECIST criteria on the basis of a CT scan. Median OS of the PP population was 18.1 months without any statistically significant difference due to sCT (20.6 without sCT vs. 18.1 without sCT, *p* = 0.53). Median PFS was 7.4 months. The objective was widely achieved, but a limitation of the study was that there could have been a heterogeneous group of patients, in terms of primary neoplasia, treated and not with sCT. It should be emphasized that no statistically significant differences were reported between the two subgroups, neither in terms of DCR, nor survival. This is another aspect that can identify PIPAC as a useful treatment in fragile patients who are no longer able to withstand systemic treatments. If on the one hand having patients within the population undergoing systemic chemotherapy in association with PIPAC can allow a comparison between the two subgroups, on the other it does not allow an optimal evaluation of PIPAC effectiveness. Furthermore, considering the small size of the sample and the heterogeneity of the pathologies, it is even more complex to analyze the results in terms of DCR.

The analysis of metabolic behavior has demonstrated a clinical tumor response rate at FDG- PET, according to PERCIST criteria (version 1.1), of 32.5%. It was noted that in many cases, FDG-PET was not specific either due to the histopathologic subtype (mucinous neoplasia, signet ring cells carcinoma) or due to inflammatory peritonitis post-PIPAC procedure.

Considering the pathologic analysis, satisfying rates of objective histological regression (22.5%) and stability (27.5%) were observed. We must point out that in some cases it was not possible to perform biopsies, or the specimens were not adequate for the assessment.

In summary, this study shows that PIPAC is active and well tolerated in patients with PC of different origins, even without additional palliative intravenous chemotherapy. On the other hand, even the combined treatment based on sCT and PIPAC is well tolerated without any significant change in terms of quality of life. The real role and timing of PIPAC in the treatment of PC has yet to be investigated. Large studies with single pathology patient cohorts are necessary to define its use and validate the results.

## Figures and Tables

**Figure 1 biomedicines-08-00102-f001:**
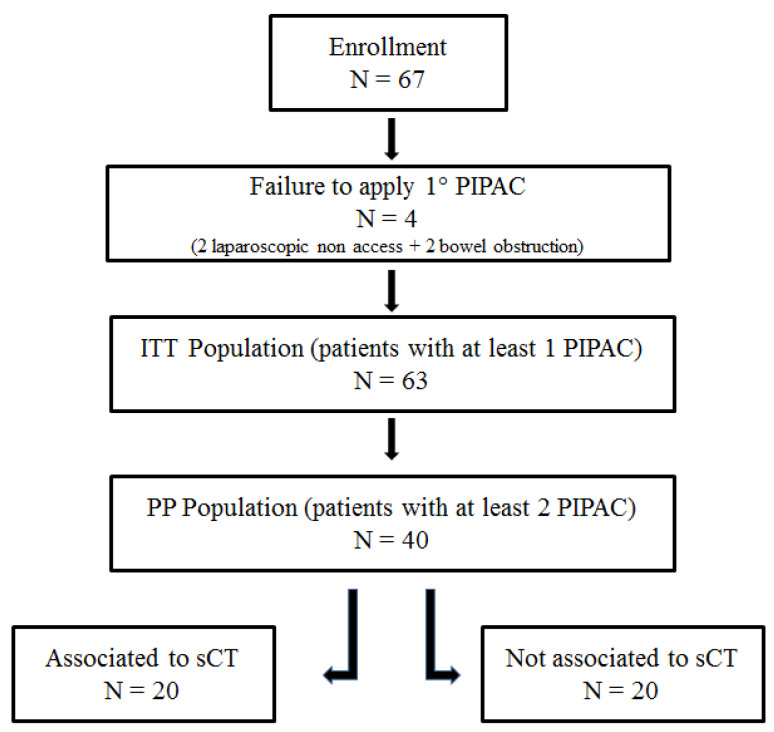
Diagram of patients’ flow through the study.

**Figure 2 biomedicines-08-00102-f002:**
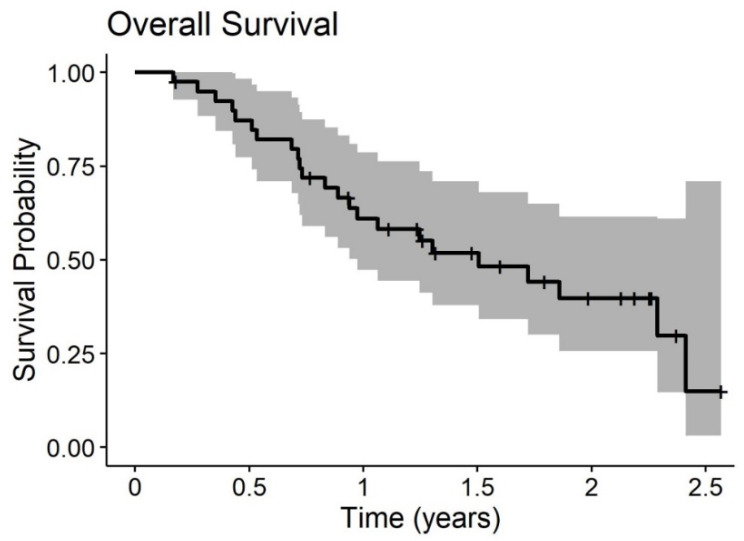
Overall survival of 20 patients with peritoneal carcinomatosis (PC) treated with at least two PIPAC procedures.

**Figure 3 biomedicines-08-00102-f003:**
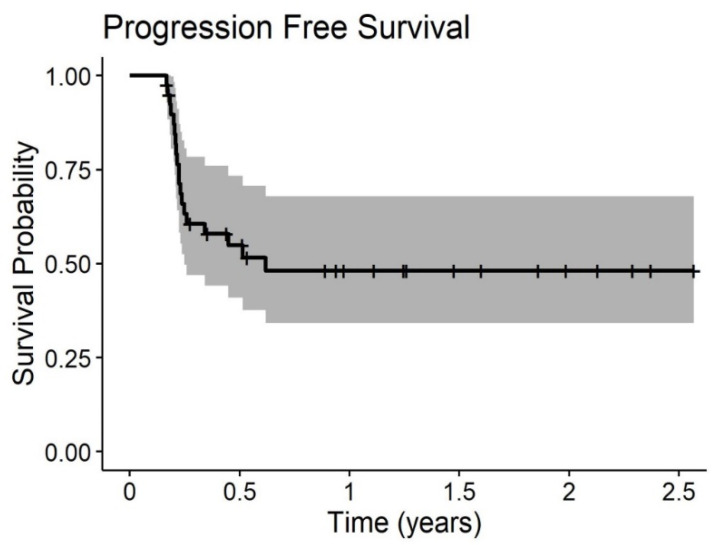
Progression-free survival of 20 patients with PC treated with at least two PIPAC procedures.

**Figure 4 biomedicines-08-00102-f004:**
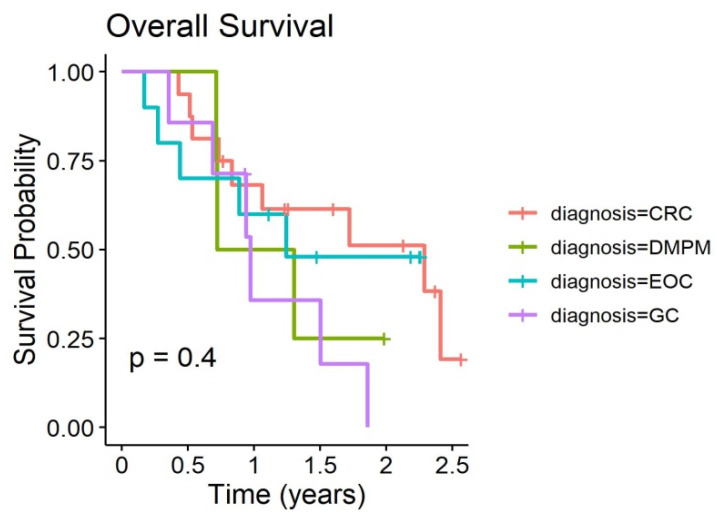
Overall Survival of 20 patients with PC stratified by pathologies treated with at least two PIPAC procedures.

**Figure 5 biomedicines-08-00102-f005:**
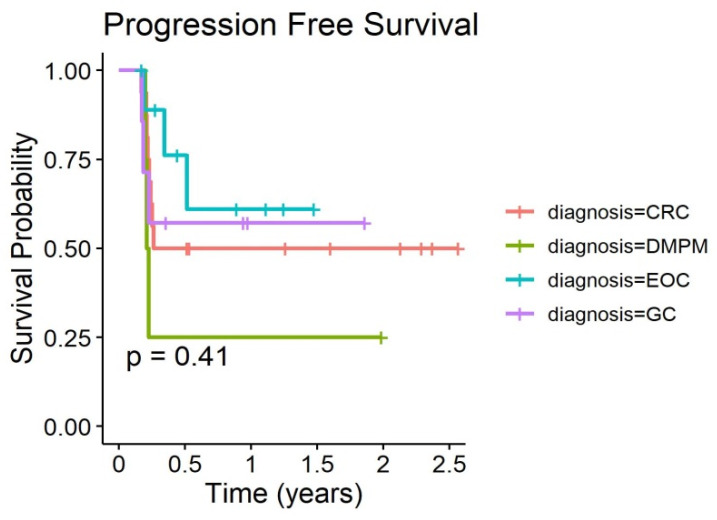
Progression-free survival of 20 patients with PC stratified by pathologies treated with at least two PIPAC procedures.

**Figure 6 biomedicines-08-00102-f006:**
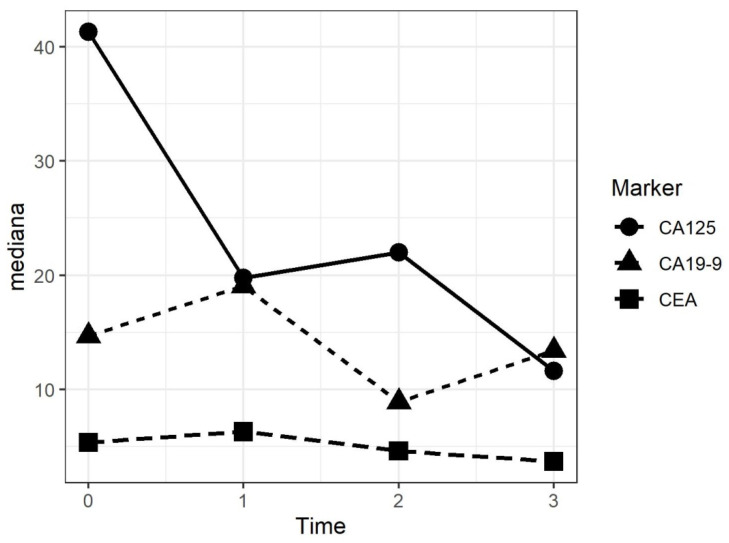
Trend of the median values of tumor markers after each PIPAC procedure.

**Figure 7 biomedicines-08-00102-f007:**
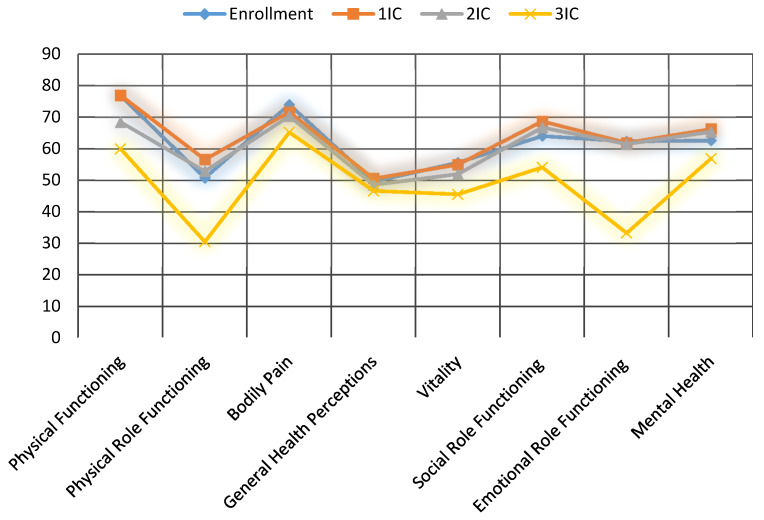
Influence of PIPAC on QoL assessed with SF-36 questionnaire.

**Table 1 biomedicines-08-00102-t001:** Patients features undergoing pressurized intraperitoneal aerosol chemotherapy (PIPAC) procedure.

Patients Characteristic	ITT Population	PP Population
Number of patients	63	40
Disease (%)		
CRC	23 (36.5%)	16 (40%)
DMPM	6 (9.5%)	4 (10%)
EOC	18 (28.6%)	10 (25%)
GC	10 (15.9%)	7 (17.5%)
PMP	4 (6.3%)	2 (5%)
PPC	2 (3.2%)	1 (2.5%)
Age (yrs; mean ± SD)	58.9 ± 10.4	60.8 ± 10.2
Male: Female	01 January 1900	00 January 1900
ECOG Performance Score (%)		
0	39 (62.9%)	21 (52.5%)
1	19 (30.6%)	15 (37.5%)
2	3 (4.8%)	3 (7.5%)
3	1 (1.6%)	1 (2.5%)
Charlson Comorbidity Index (%)		
0–2	51 (81%)	32 (80%)
3–4	10 (15.9%)	6 (15%)
>4	2 (3.1%)	2 (5%)
sCT (%)		
Yes	29 (46%)	20 (50%)
No	34 (54%)	20 (50%)
Prior Surgical Score (%)		
0	6 (9.5%)	5 (12.5%)
1	18 (28.6%)	12 (30%)
2	24 (38.1%)	16 (40%)
3	15 (23.8%)	7 (17.5%)
Serum markers (median [IQR])		
CEA	6.5 [3.05, 27.15]	5.35 [2.48, 16.57]
CA125	81 [9.85, 368.1]	41.30 [9.8, 367.5]
CA19–9	16.5 [6.18, 64.78]	14.70 [6.75, 24.5]

CRC = colorectal cancer; DMPM = diffuse malignant peritoneal mesothelioma; EOC = epithelial ovarian cancer; GC = gastric cancer; PMP = pseudomyxoma peritonei; PPC = primary peritoneal cancer.

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
