# Peer review of "Pressurized Intraperitoneal Aerosol Chemotherapy (PIPAC) with Oxaliplatin, Cisplatin, and Doxorubicin in Patients with Peritoneal Carcinomatosis: An Open-Label, Single-Arm, Phase II Clinical Trial"

_biomedicines, 2020, doi:10.3390/biomedicines8050102_

Round 1

Reviewer 1 Report

The authors submit their results of a phase II clinical trial of PIPAC with two different drug regimens for a multitude of different histologies.  The authors achieved their goal of demonstrating safety and efficacy with 35% objective response rate with a predetermined threshold of 30% as clinically significant.  I believe the manuscript is relatively well done and provides a significant contribution to the literature in this field.  I have the following recommendations for improvement

  1.  the abstract is very confusing. The authors report 82 patients in the study but only 63 were enrolled and 40 eligible for analysis. Why were over half of treated patients excluded from the study?  This needs to be very clear and transparent as it looks like the authors cherry picked their cases to analyze and thus could bias the results 
  2. in the introduction and discussion the authors fail to mention and compare their results to previous phase II studies of PIPAC

Stroller F. et al. Pressurized intraperitoneal aerosol chemotherapy with low-dose cisplatin and doxorubicin (PIPAC C/D) in patients with gastric cancer and peritoneal metastasis: a phase II study.  Ther Adv Med Oncol. 2019 May 13;11:1

  1. the results should be cleaned up a bit   Each end-point for the ITT analysis should be presented first and clearly described as ITT.  Then the authors can present each end point foe the PP analysis results and clearly describe these as PP.  p-values should be reported  not just “not statistically significant”.  
  2. Can the authors add why patients didn’t make it to the 2nd PIPAC to the flow diagram in figure 1 please.   I am assuming post progressed   This needs to be discussed in the discussion   
  3. Figure 4 and 5 were very difficult to read in black and white   Can the authors use some other method to differentiate the lines?
  4. In the methods could the authors report the time interval between PIPAC procedures?  Also what was the time interval between IV chemo and PIPAC?  Did authors alternate IV and PIPAC or IV first then PIPAC?
  5. Can the confidence interval be added to Figure 6?
  6. Was postoperative mortality defined as 30 days?
  7. Can the authors clarify the PERCIST responses?  How many had CR, PR, SD, PD or were not evaluable?
  8. Similarly, can the authors report the RECIST responses as ORR, CR, PR, SD, PD or unevaluable?

Author Response

Dear reviewer 1,

  • The different numbers refer to: 82 patients is the whole casuistry, 63 patients have characteristics that meet the protocol inclusion criteria, 40 patients were submitted to at least 1 PIPAC procedure. As suggested, I summarized the information.
  • Accepting his suggestion, I included in the discussion the phase 2 study you mentioned and another one by Tempfer. However, it must be remembered that it is difficult to compare these three studies in terms of results since ours includes multiple pathologies, that by Struller is about peritoneal carcinomatosis of gastric origin and the paper by Tempfer about the ovarian origin one.
  • Considering the request about fig. 4 and 5, I’m awaiting for the modified images by the statistician. I will attach them as soon as they are ready.

Reviewer 2 Report

Congratulations for the nice piece of work you have accomplished. Just to let you know I am not an Oncologist or Gastroenterologist, but I have been working with the peritoneal cavity/membrane for over 40 years, in the field of nephrology.

My first comment, hopefully taken as a constructive remark is that there many acronyms that one, who is not an expert in Oncology/Gastroenterology, may find hard to guess (for example, PERCIST criteria) and some others who are more "familiar", but that should be spelled ou the first time they appear in the text ( for example, ITT-Intention To Treat). As a matter of a fact, If I am not confused, there is one PP used with two different meanings (Per Protocol and in Table 1 as Primary Peritoneal Cancer in the footnote). Other acronyms that I could not find them spelled out the first time I found them in the text: ORR, IPC, OS, TTP, PFS, FDG,PRGS, ECOG. In the discussion section, line 148, you have a typo with the acronym CTACAE

I congratulate you for the novelty (Pressurized intraperitoneal Aerosol), at least for me and the intra-peritoneal use of drugs as aerosol. I think it would be very nice if you had a Picture or a Cartoon of how the Operative Procedure flows and/or looks like.

It is not clear to me (be patient with my ignorance) how the aerosol containing Cisplatin 7.5 mg/sm body surface in 150 ml NaCl 0,9% + Doxorubicin 1.5 mg/sm body surface in 50 ml NaCl 0,9% were administered as they seem to be prepared as two different "solutions/compounds". Were they given one after the other without any time interval ? 

What is the possibility of the single-port access utilized by you "to be in place" for the total period of time (18-24 weeks) the patient will be submitted to the PIPAC ? 

Would you make any comment in the discussion section, if your results would suggest that PIPAC could be evaluated earlier after the diagnosis of these pathologies ?

CA 125, CEA and CA 19-9 were measured in the blood, correct ? Have you thought of measuring in the ascites effluent as the concentration or appearance rate of cancer antigen 125 (CA125) in peritoneal dialysis (PD) effluent has been used for many years as a biomarker for mesothelial cell mass in patients on PD. In line 109, you have a typo to be corrected Ca19.9

Please read this paragraph below:

The protocol initially envisaged the enrollment of non-candidates for systemic chemotherapy (sCT); as a result of poor accrual problems, patients performing sCT were also included. Therefore, it was possible to split the population into two subgroups: patients undergoing sCT and PIPAC (n =20) and patients undergoing exclusive PIPAC procedure (n = 20). Diagram of the patients' flow through the study is depicted in Figure 1.

Is it possible to categorize each of the two sub-groups with names to be used throughout the article ? For example: sCT- associated and non-sCT-associated or sCTA and non-sCTA.

Have you had any episode of infectious peritonitis during the study ? you wrote in the discussion "It is noted that in many cases FDG-PET was not specific either due to the histopathologic subtype (mucinous neoplasia, signet ring cells carcinoma) or due to inflammatory peritonitis post-PIPAC procedure.

How do you define and identify "inflammatory peritonitis post-PIPAC procedure" ? Would an increase in the infusion volume of NaCl 0,9% or dextrose impact (positive ? negative ?) on the results obtained ?

Once again,my congratulations to the nice study and hope it can be read by many colleagues.

Author Response

Dear reviewer 2,

thanks a lot for the appreciation.

  • As for abbreviations, I proceeded to specify each acronym at the first appearance in the text.
  • Considering the administration of the drugs, it occurs in sequence (as soon as the infusion of cisplatin ends, we administer doxorubicin) and then we wait 30 min.
  • The single-port access is a way to perform laparoscopic operation using a particular platform. This platform is introduced in the abdominal wall through a very small midline incision; at the end of the operation the platform is removed. During the following operation, after removing the scar of the previous surgery, a new platform is positioned. I hope I was clear J
  • We never thought about measuring markers within the ascites; we can evaluate it in a future study.
  • The inflammatory peritonitis post-PIPAC is a chemical effect due to drugs.
  • We can not increase or reduce the infusion of NaCl 0.9% as we have to follow the approved protocol about PIPAC published by Reymond et al.

Reviewer 3 Report

This study has been designed as phase II clinical trial and concluded that PIPAC is active and well tolerated for the patients with PC of different origin even without additional systemic chemotherapy.

1. Primary endpoint of this study was overall response rate (ORR) and the hypothesis of this study was PIPAC could induce ORR in ≥ 30% patiients. However, 7 patients had a partial response (17.5%) and 7 patients a stable disease (17.5%). There was no patients in complete response (0%). Thus, ORR of this study was 17.5%, not 35%. ORR means the rate of CR + PR, not PR + SD. This phase II clinical trial did not met.

2. Enrolled patients had measurable peritoneal tumor by CT scan. This means the size of tumor must be more than 10mm. The readers cannot believe solid peritoneal tumor disappeared by only 2 cycles PIPAC. Author should present peritoneal cancer index (PCI) of patients before PIPAC and representative CT image before and after PIPAC.

Author Response

Dear reviewer 3,

  • The endpoint of the study was the overall response rate (ORR); as explained in materials and methods section, ORR was defined as the total number of patients with complete radiological response, partial response and stable disease.
  • As detailed in the results section, we didn’t report any complete radiological response (I do agree it’s strange – and pretty impossible - to see a solid peritoneal lesion disappeared). We reported stability of disease or partial response (30% reduction of the sum of the target lesion diameters)

Round 2

Reviewer 3 Report

Even if ORR was defined as the rate of CR plus PR  plus SD in Materials and Methods by author, that is defined as DCR, not ORR according to RECIST.  

Thus, this phase II study did not achieve primary endpoint.

Author Response

Dear reviewer, 

I apologize for the error in the terms; the statistical design is correct and the hypothesized result has been obtained. I mistakenly have defined ORR what actually is DCR (correctly explained in materials and methods, but incorrectly defined).
I proceeded to correct all the terms.

Thanks again